# Complete Cycle Mapping Using a Quantitative At-Home Hormone Monitoring System in Prediction of Fertile Days, Confirmation of Ovulation, and Screening for Ovulation Issues Preventing Conception

**DOI:** 10.3390/medicina58121853

**Published:** 2022-12-15

**Authors:** Andrea K. Wegrzynowicz, Amy Beckley, Aimee Eyvazzadeh, Gary Levy, John Park, Joshua Klein

**Affiliations:** 1Department of Biochemistry, University of Wisconsin-Madison, Madison, WI 53706, USA; 2MFB Fertility, Inc., 720 Austin Ave Suite 100-305, Erie, CO 80516, USA; 3Aimee Eyvazzedeh MD, Inc., 5401 Norris Canyon Road, Suite 106, San Ramon, CA 94583, USA; 4Fertility Cloud, Inc., 2100 Geng Rd, Palo Alto, CA 94303, USA; 5Carolina Conceptions, 2601 Lake Dr 301, Raleigh, NC 27607, USA; 6Extend Fertility, 200 W 57th St 1101, New York, NY 10019, USA

**Keywords:** quantitative monitoring of the menstrual cycle, estrone-3-gluconoride, pregnanediol, luteinizing hormone, follicle-stimulating hormone

## Abstract

*Background and Objectives:* To achieve pregnancy, it is highly beneficial to identify the time of ovulation as well as the greater period of fertile days during which sperm may survive leading up to ovulation. Confirming successful ovulation is also critical to accurately diagnose ovulatory disorders. Ovulation predictor kits, fertility monitors, and tracking apps are all available to assist with detecting ovulation, but often fall short. They may not detect the full fertile window, provide accurate or real-time information, or are simply expensive and impractical. Finally, few over-the-counter products provide information to women about their ovarian reserve and future fertility. Therefore, there is a need for an easy, over-the-counter, at-home quantitative hormone monitoring system that assesses ovarian reserve, predicts the entire fertile window, and can screen for ovulatory disorders. *Materials and Methods*: Proov Complete is a four-in-one at-home multihormone testing system that utilizes lateral flow assay test strips paired with the free Proov Insight App to guide testing of four hormones—FSH, E1G, LH, and PdG—across the woman’s cycle. In a pilot study, 40 women (including 16 with a fertility-related diagnosis or using fertility treatments) used Complete for one cycle. *Results*: Here, we demonstrate that Proov Complete can accurately and sensitively predict ovarian reserve, detect up to 6 fertile days and confirm if ovulation was successful, in one easy-to-use kit. Ovulation was confirmed in 38 cycles with a detectable PdG rise. An average of 5.3 fertile days (from E1G rise to PdG rise) were detected, with an average of 2.7 days prior to LH surge. Ovulation was confirmed via PdG rise an average of 2.6 days following the LH surge. While 38/40 women had a PdG rise, only 22 had a sustained PdG level above 5 μg/mL throughout the critical implantation window, indicating ovulatory dysfunction in 16 women. *Conclusions*: Proov Complete can detect the entire fertile window of up to 6 fertile days and confirm ovulation, while also providing information on ovarian reserve and guidance to clinicians and patients.

## 1. Introduction

Detecting the full fertile window, or the time during which estrogen levels are high and conception is possible, is key for those trying to achieve pregnancy or tracking ovulation for health. However, fewer than 13% of women can correctly identify ovulation [1]. Historically, it was thought that a typical, healthy menstrual cycle lasts 28 days and the fertile days fall between days 10 to 17, based on two assumptions: that a woman ovulates on cycle day (CD) 14 and that she is fertile from the time sperm may survive preceding ovulation through the lifetime of the egg following ovulation (Figure 1a) [2]. It is now known that the majority of cycles do not fall within these parameters [3,4,5]. Only a small percentage of women ovulate on CD14, even for those with regular cycles [5,6]. Yet, popular ovulation predictor kits and period-tracking apps still rely on outdated assumptions and provide frequently inaccurate fertile window predictions, and many clinicians still rely on cycle day 21 labs, assuming a day 14 ovulation [3,4,7]. Additionally, women with ovulatory dysfunctions such as polycystic ovarian syndrome (PCOS) and primary ovarian insufficiency may ovulate irregularly, late, or not at all, due to their complex hormonal patterns. Ovulatory dysfunction may account for over half of female-factor infertility cases, making it the leading cause of female infertility [8].

Age is another risk factor for female infertility. From birth to 37.5 years, the average number of eggs has already dropped from over 1 million to below 25,000 making conception more challenging as age increases and ovarian reserve decreases [9]. Measuring basal levels of follicle-stimulating hormone (FSH) as early as CD3 has been proven to help determine the remaining ovarian reserve and future fertility, and predict ovarian response, stimulation quality, and pregnancy outcomes for in-vitro fertilization (IVF) [10,11]. However, relatively few at-home fertility test kits provide information about FSH.

Due to high rates of ovulatory dysfunction and natural variability across cycles, the timing of ovulation and the fertile window varies greatly. Fertility tracking apps are often relied on to time intercourse when trying to conceive (TTC) but they are generally inaccurate and based on one-size-fits-all cycle parameters [7]. Urinary hormone testing is a more accurate way to determine the time of fertility, and several different hormones can be accurately and sensitively detected [12,13,14,15,16,17]. A peak in serum estradiol, or its urine metabolite estrone-3-glucuronide (E1G), marks the presence of a growing follicle and the opening of the fertile window. It triggers a luteinizing hormone (LH) surge that marks the time of peak fertility. However, an LH surge may not always result in the release of a mature egg and may not be present before ovulation [18]. Additionally, as LH levels commonly remain high or even rise following ovulation cycles, use of LH alone is not an ideal way to time intercourse and opening the fertile window with E1G provides a distinct advantage over LH testing alone [19]. It is also critical to confirm ovulation by testing for an increase in serum progesterone or its metabolite in urine, pregnanediol-3 glucuronide (PdG), which is also a biomarker for the beginning of the luteal phase. Several studies show that PdG is low in the first half of the cycle and only increases after ovulation [14,15,19,20,21]. In several studies, sustained and elevated PdG during the time of implantation (7–10 days past the LH surge) was correlated with 73–75% more successful pregnancies compared to the lower excretion groups, demonstrating the importance of sustained PdG in the luteal phase and early pregnancy [22,23]. In addition, a PdG level of 5 µg/mL or higher correlated with a serum progesterone level of >5 ng/mL [15]. Therefore, PdG levels in urine can be used to confirm ovulation and ovulatory function without the need for an ultrasound or multiple serum progesterone blood draws.

Traditional lateral flow immunoassays, such as those used in ovulation predictor kits and several fertility monitors, are threshold-based tests that detect a single hormone and deliver a positive or negative result, often failing to account for differences in the length and hormone concentrations of each unique cycle. Additionally, no available kit measures ovarian reserve, identifies ovulatory disorders in women, and identifies the full fertile window all in one system, reducing both cost and effort for consumers (Figure 1B). While fertility monitors that achieve some of these goals are available, they are often prohibitively expensive or time-consuming. For instance, fertility monitors may cost $125–200 initially, not including ongoing purchases of test sticks that range from $1.50–4.50 per stick [24].

To this end, MFB Fertility developed and validated the Proov Complete Testing System, the only fertility test that allows women to determine their ovarian reserve early, detect possible ovulatory dysfunction, confirm ovulation, and track their fertile and implantation windows. The Proov Complete Testing System includes 3 FSH tests, 17 MultiHormone tests that measure E1G, LH, and PdG, and an advanced lateral flow reader (LFR) integrated into the free Proov Insight smartphone application for iOS and Android users. A PDF report is generated each cycle that can be used by healthcare professionals to guide treatment and provide further information to assist clinicians in treating fertility concerns, helping patients be proactive in their own fertility care. Proov Complete currently costs $89 for an entire cycle, making the cost-per-test stick competitive and providing the distinct advantage of using a smartphone rather than requiring the purchase and maintenance of a monitor or additional hardware.

In this study, we tested the ability of Proov Complete to detect the full fertile window (up to 6 fertile days), screen for ovulation disorders, and provide information about ovarian reserve. First, the specificity, sensitivity, and reproducibility of test strips were validated. Next, an observational pilot study of 40 women (with 38 total ovulatory cycles) tested the average time from E1G rise to LH surge and LH surge to PdG rise to confirm that Proov Complete can detect up to 6 fertile days. Finally, PdG levels were tracked during the implantation window to determine ovulatory function. Time to the PdG rise also demonstrated that Proov Complete, on average, confirms ovulation sooner than other methods on the market, providing valuable and timely information to those tracking their cycles.

## 2. Materials and Methods

### 2.1. Lateral Flow Assay Design

Proov Multi-Hormone (MH) Test Strip Design: The MFB-MULTI lateral flow assay uses gold nanoparticles and buffered sample pads designed to adjust for pH and hydration levels, filter unwanted particulates, and bind contaminants in urine that may interfere with the accuracy of the test. The MH strip contains three test lines and one control line. The three test lines measure the presence of estrone-3 glucuronide (E1G), luteinizing hormone (beta subunit) (LH), and pregnanediol-3 glucuronide (PdG). LH-beta was used as LH beta is present in urine longer than intact LH. Since some women have short LH surges, this was to ensure the greatest chance of detecting LH surge in first-morning urine/once per day testing [19]. The control line detects the movement of antibodies, validating the lateral flow assay for each test strip. The MH strip is run in a dipstick format with a sample matrix of urine. As the sample travels along the membrane surface, E1G and PdG in the sample compete with their respective test lines, causing a less intense line for positive samples (competitive format). The LH in the sample is sandwiched at the test line, causing an increase in line intensity for positive samples (sandwich format).

FSH Test Strip Design: Proov Complete also includes a lateral flow assay to detect follicle stimulating hormone and assess ovarian reserve. These contain one test line, sandwiching FSH at the test line with capture antibodies and causing an increase in line intensity for positive samples (sandwich format).

### 2.2. Specificity and Sensitivity Assays

A minimum of 360 MFB MH strips were cut per lot from three accepted lots. Three technicians tested the following spiked quality control panels: LH at: 0, 5, 10, 20, 30, 40, and 50 mIU/mL; E1G at: 0, 25, 50, 75, 100, 150, and 200 ng/mL; PdG at: 0, 2.5, 5, 7.5, 10, and 15 µg/mL. Each panel was tested with 6 replicates per lot (2 per technician), repeated for three days total. Line intensity was read using a Lumos lateral flow reader and standard curves were generated. All sample acceptance ranges listed in this study have been calculated from data gathered from three accepted validation lots. Line intensity v. concentration was plotted, best-fit lines drawn (Appendix A) and R-value was calculated from best-fit lines (linear for LH and exponential for E1G and PdG). The R values were used to determine both inter and intra-assay variability. Testing with urine negative for E1G, LH, and PdG determined specificity, and values above required sensitivity were assessed for sensitivity. Required bounds for specificity and sensitivity were set as 3 standard deviations (SD) below and above the mean of the entire tested data set.

### 2.3. Study Design

Between March and July 2021, we conducted a pilot clinical study to demonstrate the performance of Proov Complete for the quantitative measurement of FSH, E1G, LH, and PdG, and to test two hypotheses: (1) that Proov Complete can confirm successful ovulation, defined as an ovulatory event producing sustained elevated PdG during the implantation window, and (2) that Proov Complete can predict up to six fertile days. The study included 40 women aged 24 to 46 years. Study participants were recruited from a database of established customers who were actively trying to conceive, not currently using hormonal contraceptives, and had tracked at least one cycle utilizing Proov Complete and the Proov Insight App. Subjects were invited to participate in this clinical trial by email and were provided with an informed consent form prior to study participation. Only subjects who provided informed consent were registered into the study. Subjects were compensated $10 for their time. Subjects using fertility treatments or having fertility-related diagnoses were included (Appendix A).

Each woman tracked a single cycle as directed by the Proov Insight App (Figure 2a), using first morning urine (FMU) and dipping the MH strip for 15 s. After waiting 10 min, urine strips were scanned with the Proov Insight app. A baseline test to measure all four hormones was directed on cycle day 5, with cycle day 1 being the first day of red menstrual bleeding flow. FSH testing was directed on cycle days 7 and 9 as well. Testing with the MH strips was then directed based on the individual’s average cycle length, starting 4–6 days before predicted ovulation, and ended upon confirmation of a PdG rise. Finally, to assess ovulatory function, users were directed to test with the MH test strips 7–10 days post-peak fertility (marked by the first day of an LH surge), to track E1G and PdG during the implantation window. After the test on 10 days post-peak, the Proov Insight Report was generated by the app and a PDF report could be downloaded by the user.

### 2.4. Insight App Test Strip Reading

The free Proov app guides the user on how to test their urine sample correctly. After waiting for 10 min, the user provides a photo of their urine test strip with the Proov app. The application server runs a script and uses machine learning specifically designed to analyze photographed images of the test strip, control for variations in camera, lighting, and operating system, check for input or output irregularities, and mathematically derive the associated hormone levels (Appendix A). The output levels are calibrated to test strips processed and photographed in a controlled environment with known hormone concentrations.

### 2.5. Data Analysis

The count of fertile days began when the Proov Insight algorithm detected a significant E1G rise for that user. The count of fertile days ended when a significant rise in PdG was detected. Peak fertility was marked when an LH value over 25 mIU/mL was detected, and the count of fertile days ended when a significant PdG rise was detected. For each cycle, times from E1G rise to LH surge and LH surge to PdG rise as described above were determined and averaged across all women who had ovulatory cycles. Ovulatory function was assessed by counting the number of days PdG levels remained over 5 µg/mL during the 7–10 days post-peak implantation window. Ovulatory dysfunction is defined as PdG levels below 5 µg/mL on two or more of these 4 testing days.

An Ovulation Score was also given. The Ovulation Score is a measure of PdG production during the implantation window. Cycles with PdG levels above 5 µg/mL for days 7–10 post-peak were given a score of 100. Scores of 20–90 were assigned for varying PdG levels during the testing window. Scores of 10 were given to cycles with no days of PdG reaching 5 µg/mL.

## 3. Results

### 3.1. Proov Complete MH Strips Are Reproducible

For acceptable reproducibility, R-values from best fit lines for the hormone-spiked panels were evaluated. Each testing day must yield a percent coefficient of variation (CV) less than 10 across all replicates for a given hormone panel, as well as total percent coefficient of variation less than 10 across all three days. All single-day and three-day % CVs were below 4. Additionally, 90% of all R-values must be over 0.9 for each hormone. At least 94% of all R-values were over 0.9, indicating acceptable reproducibility.

### 3.2. Proov Complete Is Sensitive and Specific for LH, PdG, and E1G in Urine

Required sensitivity of the Multi-Hormone strips is set by meaningful physiological concentrations of E1G, LH, and PdG, at greater than 175 ng/mL, 25 mIU/mL, and 5 µg/mL respectively. To pass sensitivity testing, at least 90% of strips must read at or above the accepted value (3 SD above the mean) for each hormone at the tested concentration (Table 1). 100% of strips passed sensitivity testing. Specificity was tested using primarily a negative control of E1G-free urine, and secondarily with 5 mIU/mL LH (representing a baseline value that might exist in urine) and PBS. To pass specificity testing, at least 90% of MultiHormone strips at these concentrations must read less than or equal to 3 SD below the mean of all strips. For E1G and PdG, 98% of strips passed specificity testing, and for LH, 93% of strips passed (Table 1).

Negative and positive samples were also used for cross-reactivity testing as part of the verification/validation testing process. This testing (Appendix A) demonstrated that Proov Complete will perform as intended in the presence of 21 interfering substances (including caffeine, glucose, and other hormones present in urine), 3 specific gravity conditions, and in urine within physiological pH range of 4–7.5.

### 3.3. Study Participants and Demographics

Forty women aged 24–46 years were recruited for the study, based on completion of one cycle of tracking with Proov Complete (Table 2). The average age was 35 years, and 50% of participants were aged 35 or older. Forty percent had a known fertility issue, including but not limited to endometriosis, PCOS, and diagnosis of unexplained infertility. Fertility treatments were used by 37.5% of women, including 8 women using ovulation induction and 9 using progesterone Appendix A. Overall, 70% of study participants had been trying to conceive for over 6 months.

### 3.4. Proov Complete Provides Information on Ovarian Reserve

In addition to MH tests, Proov Complete also includes FSH testing, which was completed by all participants. FSH testing is directed by the Proov Insight App on days 5, 7, and 9 (Figure 2a), and a plurality (46%) of included cycles detected highest FSH levels on day 7. 43% of cycles (*n* = 17) had at least one day of elevated FSH (over 10 mIU/mL), typically used to indicate low ovarian reserve, but elevated FSH was not correlated with low E1G or low PdG as measured by the Proov Ovulation Score. Four of these 17 cycles included treatment with ovulation inducing drugs. Neither Ovulation Score nor maximum PdG was significantly different between the elevated FSH group and the non-elevated (*p* = 0.40 and 0.46, respectively).

### 3.5. Proov Complete Predicts Ovulation with LH Surge

All 40 women detected an LH surge (over 25 mIU/mL) with the MH tests, ranging from days 10 to 23. For 38 of these cycles, the LH surge was followed by a PdG rise 0–6 days later, indicating the LH surge resulted in ovulation. Of the two cycles that did not result in ovulation, one had three total LH surges and was in a woman previously diagnosed with PCOS. The second ultimately saw no PdG rise after 51 days.

### 3.6. Proov Complete Confirms Ovulation with PdG Rise

Of the 40 cycles tracked, 38 were ovulatory and 2 were anovulatory (Table 3). PdG levels of all the women were low prior to ovulation, and only 38 women had increased PdG levels after ovulation. Proov Confirm PdG tests are FDA cleared to confirm successful ovulation—when PdG levels are sustained above a 5 µg/mL threshold 7–10 days after peak fertility is detected via LH. By following PdG rise detected by the algorithm after peak day that eventually resulted in successful ovulation and comparing them to baseline values, it was determined that ovulation may be confirmed with a PdG value as low as 2.5 µg/mL.

In 37 of the 38 cycles that had an increase in PdG levels after the LH surge, PdG levels reached above 5 µg/mL at least once during the 7–10 days past the LH peak period. The only woman that did not reach 5 µg/mL did achieve a level of 4 µg/mL, while her levels pre-ovulation were all under 1.5 µg/mL compared to an average of 0.81 µg/mL pre-ovulation across the cohort (calculated before LH surge).

### 3.7. An Average of 5.3 Fertile Days Are Detected

The fertile window starts the day of the E1G rise and ends the day of the PdG rise (Figure 2B). Of the 38 ovulatory cycles, the shortest fertile window was 1 day which was observed in two women. The first woman had an E1G rise and LH surge on the same day, and then she had a PdG rise the following day. The second woman had no E1G rise and a PdG rise the day following her LH surge. The longest fertile window observed was 11 days. This woman had 7 days between an E1G rise and LH surge, and a PdG rise followed 4 days after the LH surge (and no infertility diagnosis). The average fertile window was 5.32 days (Figure 2B), with an average of 2.68 days between E1G rise and LH surge, and 2.64 days from LH surge to PdG rise (Table 3). While there were slight differences in these parameters for the group using no fertility treatment and the group using treatment or having a diagnosis, these were determined to be not statistically significant.

### 3.8. Ovulatory Function

A successful ovulation is defined as when an egg is released and PdG levels remain elevated during the implantation window, 7–10 days post-peak fertility. Of the 40 cycles tracked, 22 had sustained PdG levels over 5 µg/mL for three or more of the four testing days, and thus were successful for ovulation function. Two cycles showed no increase in PdG and thus did not ovulate. The other 16 women had low PdG production during the implantation and ovulatory dysfunction. There was no correlation between ovulatory dysfunction and elevated FSH/diminished ovarian reserve.

## 4. Discussion

The Proov Complete Testing System tests four key hormones—FSH, LH, E1G, and PdG—to assess ovarian reserve, predict up to six fertile days, and confirm successful ovulation. After testing, the system generates a comprehensive report with an Ovulation Score that empowers clinicians to create effective personalized treatment plans and/or monitor the efficacy of medications or treatments administered. Patients are also able to understand the report and use it to time intercourse correctly. The Complete system is over-the-counter, non-invasive, easy-to-use, accurate and precise, confidential, and does not require the purchase of any electronic test reading devices.

The MH strips passed reproducibility, sensitivity, specificity, and cross-reactivity/interfering substance testing, often significantly above the required thresholds. Normal interfering substances and/or interfering conditions that may be present in female urine should not negatively impact this assay, and the MH strips will provide sufficient sensitivity and specificity to most users. While the pH of urine samples for best performance is limited to 4–7.5, this is consistent with healthy urine pH levels of 4.5–8. Some users at the higher end of the urine pH range may experience slightly higher E1G results than actual values, but this is unlikely to be either large in magnitude or impact the timing of intercourse for conception.

The Complete Kit also includes testing for follicle-stimulating hormone (FSH), making it the first at-home urinary hormone testing kit to do so. Previously, FSH testing would need to be at least purchased separately, and with almost all other kits required a blood sample. Measuring FSH at home, over several days, provides users with a much better understanding of their FSH levels, which fluctuate through the beginning of the cycle. As FSH levels increase as menopause approaches and ovarian reserve decreases, this is valuable information for those trying to conceive and especially considering fertility treatments but is not often included in comparable testing protocols [25].

In this study, 43% of tested cycles had elevated FSH (above 10 mIU/mL) on at least one day. Discovering elevated FSH levels in so many women in just a small study, especially as elevated FSH did not directly correlate with age, indicates that increased FSH testing may be beneficial among those trying to conceive, as elevated FSH may be more common than is known. Elevated FSH was not explained by use of ovulation inducing (OI) medications, as only 4/17 women with elevated FSH were using OI medication, and the average maximum FSH level of the OI group was lower than the non-OI elevated FSH group (11.5 vs. 15.8, respectively). The FSH results were also not correlated with PdG levels, indicating that testing ovarian reserve is separate from successful ovulation and both merit inclusion in a comprehensive hormone testing kit. Prior studies have indicated that commercial LH tests with a threshold of 25 mIU/mL detect 97–99% of ovulations [17], and the Complete MH strips were consistent, having detected 100% of the ovulations that occurred in the 40 tested cycles. It is likely that in a much larger population, some ovulations may not be detected by the LH component of the MH strip, but the inclusion of E1G testing would still help time intercourse for conception and the PdG testing would still confirm ovulation after the “missed” LH surge. Conversely, two women in our study population detected LH surges without confirming an ovulation, and with only traditional LH tests no additional information would have been provided to indicate to these women that they did not actually ovulate. The lack of a PdG rise detected by the MH strips indicated that successful ovulation had not occurred, demonstrating the benefit of MH strips and the Complete kit over LH testing alone.

Based on the 38 ovulations detected in this study, ovulation can be confirmed by Proov Complete when PdG levels reach 2.5 µg/mL or greater following an LH surge. This occurred an average of 2.64 days post-peak, which is much sooner than Proov Confirm’s previous requirement of waiting until day 7 post-peak to confirm ovulation (although the threshold of 5 µg/mL for successful ovulation is still tested days 7–10 post-peak). Having PdG data during the fertile window also allows users to continue to have intercourse and try to conceive even after the LH surge since they know that PdG may not have risen yet and ovulation may not have occurred yet. Proov Complete is the only at-home test kit that gives this PdG information with the same test strips as E1G and LH without requiring the purchase of an expensive device. Using Proov Complete to confirm ovulation is also more convenient and affordable than invasive procedures such as ultrasounds or progesterone blood tests.

Ovulation function was also assessed. While 38 women did ovulate (as measured by a rise in PdG), only 22 of these women had a successful ovulation (as defined as sustained elevated PdG 7–10 days post-peak), confirming ovulatory dysfunction may occur in approximately half of women trying to conceive [8]. Twenty-eight of the 40 women reported to have been TTC longer than six months and 27 of the women were either 35 or older and/or had a known infertility issue. This shows that confirming ovulation is not enough to help women conceive and that assessing ovulation health and function is a better way to identify causes of infertility to enable fertility treatment sooner. In addition, the Ovulation Score assists both patients and clinicians in quantifying ovulation function and providing guidance on treatment.

Our second hypothesis was that Proov Complete can predict the entire fertile window, which is up to 6 fertile days [26]. The study demonstrates that by tracking E1G, LH, and PdG in the same urine samples across the cycle, Proov MH tests detect up to 11 potentially fertile days, with the average number of fertile days being 5.32 days. While some women could experience a detected fertile window lasting 8, 9, or even 11 days, this is based on opening the fertile window with E1G rise for the greatest chance of conception. Thus, women with 8, 9, and 11 detected fertile days are not actually fertile on all of those days.

While not reflected in our study, the Proov Insight Report (Figure 3) is also an important part of the Proov Complete kit. Infertility patients, especially those who are navigating the challenges of trying to conceive, need reliable information from reputable sources [27]. The Proov Insight report provides users with information about what their hormone levels mean, in a format that is easy for both patients and their providers to interpret and discuss together. The Proov Insight App and algorithm (Appendix A) are also key in providing a seamless patient experience and allowing the Complete kit to provide high-quality hormone data without needing to purchase an expensive device. Interestingly, 15 of the 40 enrolled women were using the Complete kit to track their cycle while on OI medications including letrozole and clomiphene citrate, or luteal phase progesterone. No significant differences were observed in the fertile window tracking or ovulation scores for these women compared to the group using no fertility treatment. Therefore, Proov Complete’s results can also be used hand in hand with standard treatments. This can empower clinicians and patients to monitor effectiveness of the treatment protocol and develop personalized protocols based on hormone values to improve patient outcomes and conception rates.

## 5. Conclusions

Overall, this study demonstrated that the new Proov Complete 4-in-1 hormone test kit, including the Multi-Hormone lateral flow assay, is reproducible as well as sufficiently sensitive and specific. Proov Complete can also confirm ovulation using PdG results, as well as predict up to 6 fertile days for most users as well as detect the entire fertile window for many users. Proov Complete provides those trying to conceive and tracking their hormones with an easy-to-use, at-home test kit and Insight report, helping guide them to their goals.

## 6. Patents

Multiple patents have been issued and filed for Proov Complete and based on work presented here.

## Figures and Tables

**Figure 1 medicina-58-01853-f001:**
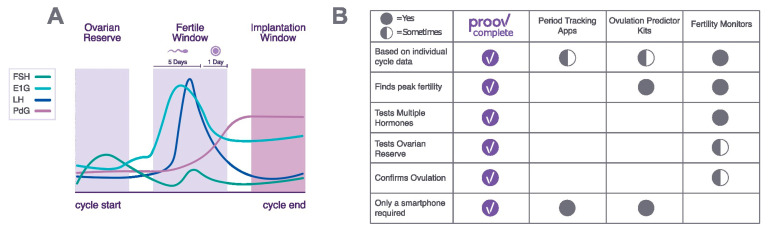
The need for a comprehensive at-home test kit to detect the entire fertile window. (**A**) Proov Complete will test FSH, E1G, LH, and PdG to find the entire fertile window (the combined time of sperm and egg survival) based on urinary hormone levels. The implantation window is also included in testing. (**B**) Period tracking apps, ovulation predictor kits, and fertility monitors are all common ways of timing intercourse for conception, but all fall short of satisfying consumer needs.

**Figure 2 medicina-58-01853-f002:**
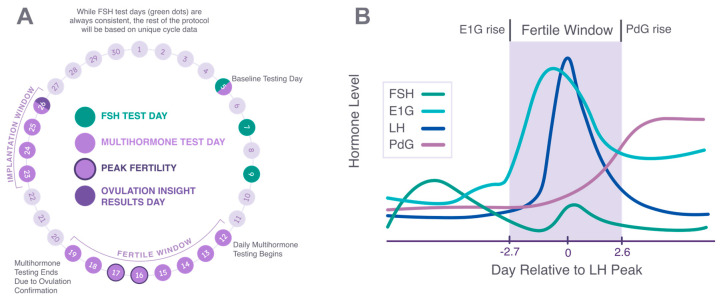
Proov Complete successfully directs testing to find up to 6 fertile days and confirm ovulation. (**A**) The Proov Insight App directs testing. FSH testing is always on days 5, 7, and 9, and fertile window testing begins according to an individual’s unique cycle data. Ovulation confirmation by PdG rise ends fertile window testing, and peak fertility based on LH surge then determines timing of implantation window PdG testing. On the final day of testing, an Insight Report is generated (see Figure 3). (**B**) Complete finds up to 6 fertile days (average of 5.32) based on E1G rise opening the fertile window and PdG rise closing the fertile window.

**Figure 3 medicina-58-01853-f003:**
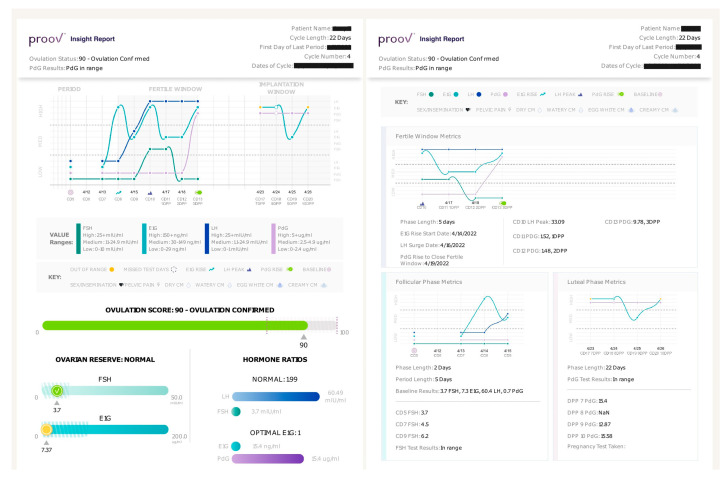
Proov Insight Report provides complete cycle mapping and guidance to patients and providers. The Proov Insight App generates a report on the final day of testing explaining hormone levels, providing an Ovulation Score, and summarizing overall hormone trends. The report also provides guidance on what the results mean and what to do next for those trying to conceive.

**Table 1 medicina-58-01853-t001:** Specificity and Sensitivity Testing.

Hormone	Test	Sample	Acceptance Range	Average Value	% of Samples Accepted
E1G	Specificity	Negative E1G Urine	>0.1221	0.1569	98.15
	Sensitivity	200 ng/mL E1G	<0.0894	0.0616	100
LH	Specificity	5 mIU/mL L.H.	<0.0765	0.0593	92.60
	Sensitivity	30 mIU/mL L.H.	>0.172	0.2588	100
PDG	Specificity	Phosphate Buffered Saline	>0.3066	0.3745	98.15
	Sensitivity	7.5 µg/mL PdG	<0.0821	0.0486	100

**Table 2 medicina-58-01853-t002:** Study Participant Characteristics.

	Average	Range
Age (years)	35	24–46
Fertility Factors	Number	Percent of Total
35 years or older	20	50
Diagnosed subfertility, using fertility treatment	8	20
Diagnosed subfertility, no treatment No diagnosis, using fertility treatment No treatment or diagnosis (Regular Cycles)	4 4 22	10 10 55
Time TTC (months)		
0–6	12	30
7–12	13	32.5
13+	15	37.5

**Table 3 medicina-58-01853-t003:** Cycle Analysis.

Total Cycles	40		
Ovulatory Cycles	38		
Anovulatory “Cycles”	2 ^1^		
Cycle Parameters (Days)	All Ovulatory Cycles	Untreated cycles, no diagnosed conditions	Cycles with fertility treatment and/or condition
Total Fertile Window	5.32	5.27 ^2^	5.12 ^2^
Range	1–11	1–11	1–9
CI (95%)	[4.65, 6.00]	[4.43, 6.24]	[4.12, 6.26]
E1G rise to LH peak	2.68	2.77 ^3^	2.38 ^3^
Range	0–7	0–7	0–5
CI (95%)	[2.06, 3.30]	[1.93, 3.69]	[1.53, 3.22]
LH peak to PdG rise	2.64	2.50 ^4^	2.81 ^4^
Range	0–7	0–7	0–6
CI (95%)	[2.16, 3.11]	[1.90, 3.15]	[2.05, 3.57]
Average # of Days with PdG > 5	2.78	2.43 ^5^	3.25 ^5^
Average Ovulation Score	70	67.4 ^6^	73.5 ^6^

^1^ Both had a PCOS diagnosis. Neither were included in further cycle analysis. ^2^ *p* = 0.84 ^3^ *p* = 0.51 ^4^ *p* = 0.58 ^5^ *p* = 0.25 ^6^ *p* = 0.56.

## Data Availability

The data presented in this study are available on request from the corresponding author.

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
