# Peer review of "Complete Cycle Mapping Using a Quantitative At-Home Hormone Monitoring System in Prediction of Fertile Days, Confirmation of Ovulation, and Screening for Ovulation Issues Preventing Conception"

_medicina, 2022, doi:10.3390/medicina58121853_

Round 1

Reviewer 1 Report

General comments:

Most of the references are older, with only 7 of the 23 references published within the last 5 years.  There is not an excessive number of self-citations, but one citation of one of the authors #19 – Beckley et al did not include the publication date.

Here are additional references to consider:

Symul L, Wac K, Hillard P, Salathé M. Assessment of menstrual health status and evolution through mobile apps for fertility awareness. NPJ Digit Med. (2019) 2:64. doi: 10.1038/s41746-019-0139-4

Tamrakar SR, Bastakoti R. Determinants of Infertility in Couples. J Nepal Health Res Counc. 2019 Apr 28;17(1):85-89. doi: 10.33314/jnhrc.1827. PMID: 31110383.

Crawford NM, Steiner AZ. Age-related infertility. Obstet Gynecol Clin North Am. 2015 Mar;42(1):15-25. doi: 10.1016/j.ogc.2014.09.005. Epub 2014 Dec 5. PMID: 25681837.

It would have been helpful to test the device and describe the findings in clearly distinguishable groups, one group of women with regular cycles between 26-32 days who are not using any medications that may affect hormones, eg/ ovulation induction medications, progesterone, etc and then another group of women who may have underlying cycle abnormalities who may also be taking medications that can affect their hormones.

For figures 1 and 2, it would be helpful to include the hormone levels along the Y axis. Figure 3 is very blurry. Can the authors provider a larger, higher quality image? In Table 2, it would be helpful to clarify the overlap between those women who had a known fertility issue and those using fertility treatments.  That is, are all 15 women who are using fertility treatment have a known fertility issue?  It would also be interesting to know what are some examples of the fertility issues women have.  In Table 3, it would be good to state how many cycles had a fertile window greater than 6 days, since that is considered the normal length of the fertile window.

Specific comments:

Line 17 – I have never seen a reference citation in an abstract.  Also this reference is more than 30 years old.   Please find a more updated reference

Line 40 “Clinical guidelines state that a typical cycle lasts 28 days…” again the reference for this sentence is from 1998.  Are there updated guidelines highlighting that healthy cycles range from 26-32 days?

Line 55 – “Only a small percentage of women ovulate on CD 14” – again use a more up to date reference.  Recommend Symul (2019) below

Line 48 – “Ovulatory dysfunction accounts for 25% of all known causes of infertility” – again reference is 30 years old.   Consider Tamrakar (2019)

Line 58 – “as age increases and ovarian reserve decreases” – again please use updated references.  Consider Crawford (2015)

Line 66-67 – “Urinary hormone testing is a more accurate way to determine time of fertility” – again this needs a reference.  Check articles by Fehring / Marquette

Line 96- ovulatory dysfunctions – should be dysfunction

Line 97 – Delete “all in the comfort of their home” – sounds like a sales pitch

Line 101 – change the word providers to professionals

Lines 107-109 – Sentence is confusing as it is written in passive voice.  Rewrite

Line 164 – “testing … started on a day based on the individual’s average cycle length”. Please be more specific about how the start day is determined

Line 217 – please spell out the abbreviation PBS as it is not clear what this refers to

Lines 232-33 – since 8 women used ovulation induction and 9 used progesterone supplementation, how can you draw conclusions about general rates of ovulatory dysfunction?

Line 240 and 242 – 43% and 4 should be written out as words since thyey begin sentences

Line 241-2 – FSH was not correlated with low progesterone, but was it also correlated with low estradiol?

Line 271 – “longest fertile window observed was 11 days” – did this woman have PCOS

Line 290 – what are “fertility providers”?  clinicians would be a better word

Line 335-7 – the phrase in parentheses “(although the threshold …post peak)” might be better if moved to Line 341 just before the sentence beginning with Proov Complete

Line 346 – “38 women did ovulate” can you clarify for the 9 women who used progesterone supplementation, did they start it after ovulation was confirmed?

Line 348 – “elevated PdG after ovulation” please clarify if it is Days 7-10

Line 367 – change the word providers to clinicians

Author Response

General comments:

Most of the references are older, with only 7 of the 23 references published within the last 5 years.  There is not an excessive number of self-citations, but one citation of one of the authors #19 – Beckley et al did not include the publication date. This was a typo and we have included the publication date.

Here are additional references to consider:

Symul L, Wac K, Hillard P, Salathé M. Assessment of menstrual health status and evolution through mobile apps for fertility awareness. NPJ Digit Med. (2019) 2:64. doi: 10.1038/s41746-019-0139-4

Tamrakar SR, Bastakoti R. Determinants of Infertility in Couples. J Nepal Health Res Counc. 2019 Apr 28;17(1):85-89. doi: 10.33314/jnhrc.1827. PMID: 31110383.

Crawford NM, Steiner AZ. Age-related infertility. Obstet Gynecol Clin North Am. 2015 Mar;42(1):15-25. doi: 10.1016/j.ogc.2014.09.005. Epub 2014 Dec 5. PMID: 25681837.

Thank you! We appreciate the recommendations and have included these, as well as additional up to date references.

It would have been helpful to test the device and describe the findings in clearly distinguishable groups, one group of women with regular cycles between 26-32 days who are not using any medications that may affect hormones, eg/ ovulation induction medications, progesterone, etc and then another group of women who may have underlying cycle abnormalities who may also be taking medications that can affect their hormones.

We agree and have amended table 3 to include our separate analysis of those cycles that were regular/not using any fertility treatment or have underlying cycle abnormalities, and those cycles that were untreated. These differences were not found to be statistically significant to 95% confidence.

For figures 1 and 2, it would be helpful to include the hormone levels along the Y axis.

While we appreciate the suggestion, these figures are meant to qualitatively illustrate which hormones are highest during each phase of the cycle. As the different hormones have different absolute values both from other hormones and between individuals, we are not sure that including values on the Y-axis would be feasible or add clarity.

 Figure 3 is very blurry. Can the authors provider a larger, higher quality image?

Yes! Our apologies.

In Table 2, it would be helpful to clarify the overlap between those women who had a known fertility issue and those using fertility treatments.  That is, are all 15 women who are using fertility treatment have a known fertility issue?  It would also be interesting to know what are some examples of the fertility issues women have.  In Table 3, it would be good to state how many cycles had a fertile window greater than 6 days, since that is considered the normal length of the fertile window.

We have updated table 2 to clarify the different groups of treated vs. untreated cycles included in this study. We have also added supplemental table S2 to provide more insight on the treatments used and diagnoses of the women in the study.

Specific comments:

Line 17 – I have never seen a reference citation in an abstract.  Also this reference is more than 30 years old.   Please find a more updated reference

We have removed the reference.

Line 40 “Clinical guidelines state that a typical cycle lasts 28 days…” again the reference for this sentence is from 1998.  Are there updated guidelines highlighting that healthy cycles range from 26-32 days?

While it has been challenging to find consistency in guidelines, we have found some recommendations that regular cycles be within this range. We have updated this section to more thoughtfully reflect current guidelines as well as popularity of period-tracking apps. 

Line 55 – “Only a small percentage of women ovulate on CD 14” – again use a more up to date reference.  Recommend Symul (2019) below

Thank you. This has been changed.

Line 48 – “Ovulatory dysfunction accounts for 25% of all known causes of infertility” – again reference is 30 years old.   Consider Tamrakar (2019)

Thank you! We have edited accordingly.

Line 58 – “as age increases and ovarian reserve decreases” – again please use updated references.  Consider Crawford (2015)

This has been updated.

Line 66-67 – “Urinary hormone testing is a more accurate way to determine time of fertility” – again this needs a reference.  Check articles by Fehring / Marquette

We have added references.

Line 96- ovulatory dysfunctions – should be dysfunction

Line 97 – Delete “all in the comfort of their home” – sounds like a sales pitch

Line 101 – change the word providers to professionals

Lines 107-109 – Sentence is confusing as it is written in passive voice.  Rewrite

Line 164 – “testing … started on a day based on the individual’s average cycle length”. Please be more specific about how the start day is determined

We have specified that testing is started based on an individual’s cycle length, 4-6 days before predicted ovulation.

Line 217 – please spell out the abbreviation PBS as it is not clear what this refers to

All above edits were made.

Lines 232-33 – since 8 women used ovulation induction and 9 used progesterone supplementation, how can you draw conclusions about general rates of ovulatory dysfunction?

While this study did not attempt to determine general rates of ovulatory dysfunction, we found that 16 women did not have successful ovulation. Although the group using fertility treatments did have a greater average number of days with PdG over 5 ug/mL, this was determined to be not statistically significant from the non-treated group. Regardless of whether the unsuccessful ovulations came from the treated or non-treated groups, 16/40 is certainly high and within the realm of 50% of infertility cases. The use of fertility treatments would only obscure further possible cases of ovulatory dysfunction, which is why we did not attempt to quantify general rates.

Line 240 and 242 – 43% and 4 should be written out as words since thyey begin sentences

Line 241-2 – FSH was not correlated with low progesterone, but was it also correlated with low estradiol?

It was not. We have added this.

Line 271 – “longest fertile window observed was 11 days” – did this woman have PCOS

Interestingly, no (or at least not diagnosed). We have added this to the text.

Line 290 – what are “fertility providers”?  clinicians would be a better word

Line 335-7 – the phrase in parentheses “(although the threshold …post peak)” might be better if moved to Line 341 just before the sentence beginning with Proov Complete

Line 346 – “38 women did ovulate” can you clarify for the 9 women who used progesterone supplementation, did they start it after ovulation was confirmed?

They did. We have clarified.

Line 348 – “elevated PdG after ovulation” please clarify if it is Days 7-10

 Clarified.

Line 367 – change the word providers to clinicians

Thank you! We have made all the above edits.

Reviewer 2 Report

A few typos found:

Abstract:

Line 27 to. 28 

Doesn’t make sense.= “In a study of 40 women 27 over one cycle,”  add "provided one". Only one cycle was requested of the participants.

The abstract should make reference to the population of women that participated and that these were not all women determined to be healthy one had PCOS and another with unexplained infertility.  

The abstract is key to the integrity of the article and great care should be taken to make sure the reader understands exactly what to expect.  

Line 232 - space needed between - and9

Although, 40 participants is sufficient this is a small study and should be considered a "pilot" study.  This would improve its credibility since it might also be considered an advertisement for Proov.  

The use of 3SD for the FW is quite large given the SD was 2-17 days.  It would be good to see what the 95% CI is instead.  

Table 3 - Would be clearer if it was broken down into the results showing what the levels were in the two that were anovulatory.  Mixing these two is not helpful and may affect results.

Figure 3 - Maybe it is my computer screen but I was not able to clearly see the values, when I enlarged it the values became even more pixelated. 

Finally, cost of the product would be beneficial.  If the cost of Proov complete is less than other fertility devices it should be clearly stated and even a comparison between different devices would be appreciated.  

Author Response

A few typos found:

Abstract:

Line 27 to. 28 

Doesn’t make sense.= “In a study of 40 women 27 over one cycle,”  add "provided one". Only one cycle was requested of the participants.

The abstract should make reference to the population of women that participated and that these were not all women determined to be healthy one had PCOS and another with unexplained infertility.  

The abstract is key to the integrity of the article and great care should be taken to make sure the reader understands exactly what to expect.  

Thank you! We have adjusted the abstract accordingly.

Line 232 - space needed between - and9

Although, 40 participants is sufficient this is a small study and should be considered a "pilot" study.  This would improve its credibility since it might also be considered an advertisement for Proov.  

We agree and have edited the abstract and a few other portions of the manuscript accordingly.

The use of 3SD for the FW is quite large given the SD was 2-17 days.  It would be good to see what the 95% CI is instead.  

We appreciate the suggestion and agree! We have adjusted table 3 into include 95% CIs instead.

Table 3 - Would be clearer if it was broken down into the results showing what the levels were in the two that were anovulatory.  Mixing these two is not helpful and may affect results.

Neither of these cycles were included in further analysis. We have clarified this in the table. 

Figure 3 - Maybe it is my computer screen but I was not able to clearly see the values, when I enlarged it the values became even more pixelated. 

We have updated figure 3 with a higher-resolution version.

Finally, cost of the product would be beneficial.  If the cost of Proov complete is less than other fertility devices it should be clearly stated and even a comparison between different devices would be appreciated.  

While the cost of different devices is difficult to compare directly based on different pricing structures and individual cycle fluctuations, we have done our best and added some cost information on Proov Complete and estimates for other products.

Round 2

Reviewer 1 Report

Overall this paper is significantly improved. Thank you for incorporating many of my suggestions, including updating references and adding to the tables.  I have made a few more suggestions that you can hopefully incorporate

Line 17: I would not include a citation in the abstract, especially since you removed the fact references.  Besides citation 1 is 30 years old, so best not to include it at all

Line 41-42: The edit to the beginning of the sentence is good, but I would leave the rest of the sentence as it was originally written, so it readsL. Historically, it was thought that a typical, healthy menstrual cycle lasts 28 day and the fertile days fall between 10 to 17, ....

Line 46-47:  Although citation #6 highlight the fact that most menstrual cycles do not fall with tin the previous parameters, it is not evident that clinical guidelines have been updated, so delete the phrase, "and in many cases clinical guidelines have been updated accordingly" 

Line 50: GREAT added phrase, "many clinic rely on Cycle day 21 labs, assuming a day 14 ovulation" but would it be better to use the word clinicians, instead of clinics?

Line 100-101.  It is great that you added in information about the cost of the fertility monitors/ test strips, but it would be good to include a citation or two, such as the following:

Duane M, Martinez V, Berry M, Manhart M. Evaluation of a Fertility Awareness-Based Shared Decision-Making Tool -Part 1: Study Design and Impact on Clinician Knowledge. PEC Innovation. Dec 2022; 1. doi.org/10.1016/j.pecinn.2022.100061

Lines 256-57, for clarity rewrite as follows: "elevated FSH was not correlated with low PdG, nor E1G, as measure by the Proov Ovulation Score

Line 368 - add in the following phrase at the end of that second sentence:  While 38 women did ovulate .... "confirming ovulatory dysfunction may occur in over half of all women trying to conceive."

Author Response

See also: track changes in attached file

Line 17: I would not include a citation in the abstract, especially since you removed the fact references.  Besides citation 1 is 30 years old, so best not to include it at all

Thank you for catching this! This was an error. We did not intend to leave it in.

Line 41-42: The edit to the beginning of the sentence is good, but I would leave the rest of the sentence as it was originally written, so it readsL. Historically, it was thought that a typical, healthy menstrual cycle lasts 28 day and the fertile days fall between 10 to 17, ....

We have edited this sentence.

Line 46-47:  Although citation #6 highlight the fact that most menstrual cycles do not fall with tin the previous parameters, it is not evident that clinical guidelines have been updated, so delete the phrase, "and in many cases clinical guidelines have been updated accordingly" 

We agree-it’s been difficult to figure out what the clinical guidelines even are! We have updated accordingly.

Line 50: GREAT added phrase, "many clinic rely on Cycle day 21 labs, assuming a day 14 ovulation" but would it be better to use the word clinicians, instead of clinics?

Edited.

Line 100-101.  It is great that you added in information about the cost of the fertility monitors/ test strips, but it would be good to include a citation or two, such as the following:

Duane M, Martinez V, Berry M, Manhart M. Evaluation of a Fertility Awareness-Based Shared Decision-Making Tool -Part 1: Study Design and Impact on Clinician Knowledge. PEC Innovation. Dec 2022; 1. doi.org/10.1016/j.pecinn.2022.100061

Thank you! This is an appropriate citation.

Lines 256-57, for clarity rewrite as follows: "elevated FSH was not correlated with low PdG, nor E1G, as measure by the Proov Ovulation Score

We have rewritten to clarify.

Line 368 - add in the following phrase at the end of that second sentence:  While 38 women did ovulate .... "confirming ovulatory dysfunction may occur in over half of all women trying to conceive."

We have added this, using “approximately half” to more accurately reflect our own results.
